# Evaluation of Plant Ceramide Species-Induced Exosome Release from Neuronal Cells and Exosome Loading Using Deuterium Chemistry

**DOI:** 10.3390/ijms231810751

**Published:** 2022-09-15

**Authors:** Yuta Murai, Takumi Honda, Kohei Yuyama, Daisuke Mikami, Koichi Eguchi, Yuichi Ukawa, Seigo Usuki, Yasuyuki Igarashi, Kenji Monde

**Affiliations:** 1Graduate School of Life Science, Hokkaido University, Kita 21 Nishi 11, Sapporo 001-0021, Japan; 2Faculty of Advanced Life Science, Hokkaido University, Kita 21 Nishi 11, Sapporo 001-0021, Japan; 3Lipid Biofunction Section, Faculty of Advanced Life Science, Hokkaido University, Kita 21, Nishi 11, Sapporo 001-0021, Japan; 4Innovation and Business Development Headquarters, Daicel Corporation, Niigata 944-8550, Japan; 5Healthcare SBU Business Strategy, Daicel Corporation, Tokyo 108-8259, Japan

**Keywords:** plant ceramide, exosome, amyloid-β, Alzheimer’s disease, lipidomics, deuterium

## Abstract

The extracellular accumulation of aggregated amyloid-β (Aβ) in the brain leads to the early pathology of Alzheimer’s disease (AD). The administration of exogenous plant-type ceramides into AD model mice can promote the release of neuronal exosomes, a subtype of extracellular vesicles, that can mediate Aβ clearance. In vitro studies showed that the length of fatty acids in mammalian-type ceramides is crucial for promoting neuronal exosome release. Therefore, investigating the structures of plant ceramides is important for evaluating the potential in releasing exosomes to remove Aβ. In this study, we assessed plant ceramide species with D-*erythro*-(4*E*,8*Z*)-sphingadienine and D-*erythro*-(8*Z*)-phytosphingenine as sphingoid bases that differ from mammalian-type species. Some plant ceramides were more effective than mammalian ceramides at stimulating exosome release. In addition, using deuterium chemistry-based lipidomics, most exogenous plant ceramides were confirmed to be derived from exosomes. These results suggest that the ceramide-dependent upregulation of exosome release may promote the release of exogenous ceramides from cells, and plant ceramides with long-chain fatty acids can effectively release neuronal exosomes and prevent AD pathology.

## 1. Introduction

Alzheimer’s disease (AD) is the most common neurocognitive disorder and is a progressive neurodegenerative disorder that affects more than 50 million people worldwide. Senile plaques of aggregated amyloid-β (Aβ), produced from amyloid precursor protein (APP) following consecutive cleavage reactions by β- and γ-secretases, accumulate in brains, and this is the initial pathological stage of AD [1,2]. Therefore, drugs that inhibit Aβ processing, such as inhibitors of β- and γ-secretases, have been developed. However, despite several decades of drug discovery research and clinical trials, progress in AD treatments has advanced slowly. Additionally, since Aβ deposition begins long before the onset of cognitive deficits, precisely diagnosing early AD pathology is difficult. It is reported that functional lipid ingestion in this preclinical term may be an effective strategy for AD prevention. For instance, epidemiological research has linked the consumption of high omega-3 fatty acids such as docosahexaenoic acid (DHA) and eicosatetraenoic acid (EPA) with a lower risk of AD [3]. Our recent study demonstrated that the oral administration of glucosylceramides (GlcCer), extracted from the konjac root, into AD model mice attenuated the Aβ burden in the brain and improved cognitive activities [4]. Additionally, the oral intake of plant GlcCer for six months lowered blood biomarkers and alleviated brain amyloid burden in human subjects with healthy and mild cognitive impairment (MCI) [5].

Exosomes, a type of small extracellular vesicles, are derived from the endosomal membrane. Our previous reports showed that exosomes released from cultured neurons possess the potential for mediating Aβ clearance [4,6,7,8,9]. Neuronal exosomes associate with Aβ through their surface glycosphingolipids and are incorporated into microglia for degradation. In addition, we reported that exosome release from SH-SY5Y cells is promoted by treatment with exogenous mammalian-type ceramides (Cers), and this involves the lysosome-associated protein transmembrane 4B (LAPTM4B) [10]. Importantly, the length of the acyl chain influences activity; Cers consisting of long acyl chains (C16:0 and C18:0) enhance exosome release, whereas those with shorter and longer chains had no effects. Therefore, the structural characteristics of Cer species may be a crucial factor in promoting exosome release. However, it remains to be explored how different plant Cer species induce exosome production. In plant tissues (rice, corn, and konjac root), most sphingoid bases exist as D-*erythro*-(4*E*,8*Z* or 8*E*)-sphingadienine (d18:2) or D-*erythro*-(8*Z*)-phytosphingenine (t18:1), while most mammalian Cers (Figure 1) exist as D-*erythro*-sphingosine (d18:1). Focusing on the structures of plant Cers could reveal crucial aspects that explain their pharmacological activities. In the present work, we found that plant Cers, including D-*erythro*-(4*E*,8*Z* or 8*E*)-sphingadienine, accelerated exosome release more effectively than mammalian-type Cers that were linked to d18:1. Additionally, structural differences in the fatty acyl chains of plant Cers affected the exosome’s release; the C18 acyl chains most strongly stimulated exosome release, whereas geometric isomerism at the C8–C9 position had no effect. Moreover, tracking using liquid chromatography mass spectrometry (LC-MS) with penta-deuterium-labeled plant Cers revealed that exogenous Cers were derived from exosomes. 

## 2. Results

### 2.1. Identification of Plant Cer Types Promoting Exosome Release

One of the major classes of sphingolipids in the human diet is the plant GlcCer [11]. In this study, we selected eight major species of plant GlcCer isolated from konjac (Amorphophallus konjac) and purified them by enzymatic glucose cleavage according to a previously reported method [12]. As shown in Figure 2, no changes in cell viability were observed in SH-SY5Y cells after the treatment with each of the eight Cer species, up to 50 μM. 

Subsequently, to identify plant Cer types potentially inducing exosome production, we tested each plant Cer species and mammalian-type Cer (d18:1/18:0) using SH-SY5Y cells at a concentration of 10 μM for 24 h. Exosome release was measured by the exosome sandwich enzyme-linked immunosorbent assay (ELISA) system with T-cell immunoglobulin and mucin domain-containing protein 4 (TIM4) phosphatidylserine (PS)-binding protein, and antibody against CD63, an exosome marker protein [13]. As with mammalian Cers, plant Cer species consisting of sphingadienine and long-chain fatty acids (d18:2/16 h:0, 18 h:0) markedly increased exosome release, whereas those with sphingadienine and very long-chain fatty acids (d18:2/20 h:0, 22 h:0) and phytosphinganine (t18:1/22 h:0, 24 h:0) did not have an effect (Figure 3a). Additionally, changing cis- and trans-isomers at the C8–C9 position scarcely influenced the activity. With C18 Cers, which displayed maximum activity, the effect of plant Cers was stronger than that of mammalian Cers (Figure 3a). Consistent with the ELISA results, an analysis with a nanoparticle analyzer also revealed that plant Cers with long-chain fatty acids (d18:2/16 h:0, 18 h:0) stimulated exosome release (Figure 3b). The size of exosome particles was not altered by Cer treatment; the diameter remained at 60–140 nm, with a peak at ~100 nm. In addition to the results of SH-SY5Y, a neuronal cell line, Cers consisting of sphingadienine and long-chain fatty acids (d18:2/18 h:0) increased exosome release from primary mouse neurons and human iPS neurons (Figure 3c,d, Appendix A). The effects of plant Cers (d18:2/18 h:0) were more potent than mammalian Cers (d18:1/18:0) in both cells. 

We previously demonstrated that neuronal exosome-bound Aβ is taken up by microglia, transported through the endocytic pathway, and degraded in lysosomes [6]. To determine whether the increase in EVs induced by Cers d18:2/18 h:0 treatment promotes Aβ clearance, a transwell culture system, in which exosomes and Aβ secreted from Aβ-overexpressing SH-SY5Y cells placed on upper inserts can interact with microglial BV-2 cells placed at the bottom of the wells, was utilized. Under this experimental setting, we added plant or mammalian Cer species to the cultures at ten µM, and after 24 h of co-incubation, the levels of Aβ in the medium were determined. The levels of both Aβ40 and Aβ42 in the culture media significantly decreased following Cer d18:2/18 h:0 and Cer d18:1/18:0 treatment, but not Cer d18:2/20 h:0 and Cer d18:2/24 h:0 (Figure 4). In addition, the concentrations of Aβ40, a major species of Aβ, were much lower after the treatment with Cer d18:2/18 h:0 than Cer d18:1/18:0 treatment, suggesting that plant Cers have high potency relative to exosome-dependent clearance extracellular Aβ. 

Exogenous mammalian-type Cers promote exosome release through LAPTM4B, transmembrane proteins expressed in late endosomes and lysosomes [10]. Previous studies reported that synthetic d18:1-type Cers bind to recombinant LAPTM4B protein. To determine whether LAPTM4B is also involved in plant Cers-dependent exosome release, we performed a siRNA-mediated knockdown study. LAPTM4B knockdown in SH-SY5Y cells completely prevented the increase in exosome release induced by d18:2/18 h:0, as well as d18:1/18:0 (Figure 5a). A protein–ceramide overlay assay using an NH_2_-terminal glutathione S-transferase (GST) fusion LAPTM4B protein revealed that apparent binding signals were obtained for d18:2 4*E*,8*Z*/18 h:0 (Figure 5b). The signal densities were higher than those for d18:1/18:0, suggesting an intense association between plant Cers and LAPTM4B (Figure 5c). 

### 2.2. Analysis of the Loading of Exogenous Plant Cers into Exosomes Using Deuterium Chemistry and Mass Spectrometry

For intra- and extracellular tracing and quantification of extremely small amounts of bioactive compounds, using a combination of deuterium-labeled compounds and mass spectroscopy is an efficient strategy. Recently, we reported deuterated D-*erythro*-(4*E*,8*Z*)-sphingadienine-*d*_5_ for rapid analysis in a sphingolipidomics study [14]. Using D-*erythro*-(4*E*,8*Z*)-sphingadienine-*d*_5_, we prepared deuterated plant Cer-*d*_5_ (4*E*,8*Z* d18:2-*d*_5_/18:0, Figure 6a). Cell viability and exosome release activity of plant Cer-*d*_5_ (Figure 6b,c) were similar to unlabeled plant Cers (4*E*,8*Z* d18:2/18 h:0). 

Next, after the treatment of cells with or without deuterated plant Cer-*d*_5_ for 24 h, we extracted sphingolipids from the cell and exosome samples and measured the amount of plant Cer-*d*_5_ by LC-MS/MS (Figure 7). Extracted ion chromatograms (XIC) and MS/MS spectra of deuterated plant Cer-*d*_5_ (4*E*,8*Z* d18:2-*d*_5_/18:0) are shown in Figure 7a,b. Quantitative analyses of plant Cer-*d*_5_ present in each sample based on product ions of the 4*E*,8*Z*-d18:2-*d*_5_ moiety specifically showed quantitative amounts of sphingolipids derived from exogenous deuterated plant Cer-*d*_5_ (Figure 7c). Deuterated plant Cer-*d*_5_ was detected in exosomes, and the amount of plant Cer-*d*_5_ in exosomes was approximately three times greater than in cells, indicating that exogenously added plant Cers were incorporated into cells and released extracellularly in combination with exosomes. 

## 3. Discussion

In this study, we investigated the abilities of plant Cer species to stimulate exosome release from SH-SY5Y cells. The ability of plant Cers to induce exosome release varied with structural differences. Regarding fatty acid components, Cer species with long-chain fatty acids (C16 and C18) strongly promoted exosome release, whereas those with very long-chain fatty acids (C20, C22, and C24) did not exert an effect. Regarding sphingoid base components, plant-type moieties stimulated exosome release more effectively than mammalian-type components (d18:1/18:0 vs. d18:2/18 h:0, Figure 3a). According to the correlation between exosome release and the Cer-LAPTM4B interaction [10,15,16], the length of the fatty acid is the most significant element for Cer to bind its interaction motif in LAPTM4B; additionally, the plant sphingoid base may possess a higher affinity for LAPTM4B than mammalian sphingosine and/or assist protein–protein interactions between LAPTM4B and other downstream signaling agents, thereby inducing exosome production. Therefore, LAPTM4B could be considered to recognize the molecular structure of Cer for its binding. The most common sphingoid bases in mammals and plants have two chiral centers at the C-2 and C-3 positions. Theoretically, there are four stereoisomers, D-*erythro*, L-*erythro*, D-*threo*, and L-*threo*, in the sphingoid base, but only the D-*erythro* type has been found. In fact, L-*threo* type ceramides were reported to exhibit a higher affinity towards a lipid metabolism enzyme compared to the D-*erythro* type [17]. Thus, focusing on the stereochemistry of sphingoid bases becomes one of the fascinating studies for developing higher potential materials to promote exosome release through LAPTM4B.

In the present study, we also prepared deuterated plant Cer-*d*_5_ (4*E*,8*Z* d18:2-*d*_5_/18:0) to investigate its localization in SH-SY5Y cells, and LC-MS results revealed that exogenously added plant Cers were ~3-fold enriched in exosomes compared with cells at 24 h after treatment. We previously demonstrated that Cer species are directed to endosomal compartments, such as multivesicular bodies, and transported for recycling and exosome release through LAPTM4B [10]. In addition, LAPTM4B is primarily localized to late endosomes and lysosomes [18,19]. Therefore, this result indicates that vesicle trafficking and exosomal efflux may be triggered by the interaction of Cers with LAPTM4B localized mainly in late endosomes.

Finally, the intake of plant sphingolipids from daily food and natural sources can lead to skin barrier improvement, such as decreased transepidermal water loss and enhanced stratum corneum flexibility [20]. GlcCer is hydrolyzed into its components (glucose, a fatty acid, and a sphingoid base) by intestinal digestive enzymes for uptake by intestinal enterocytes [21]. Some sphingoid bases, including those of plant origin, are then resynthesized to Cers, GlcCers, and other complex sphingolipids, such as sphingomyelins, and then absorbed into the body [22,23,24]. Our previous study demonstrated that synthetic plant-type Cers could permeate into brains through the blood–brain barrier (BBB) in mouse and BBB cell culture models [25]. Therefore, the oral intake of plant sphingolipids and sphingoid bases as dietary supplements could help prevent AD by enhancing neuronal exosome release. There are various sphingoid bases in nature, such as sphinganine, phytosphingosine, 9-methyl sphingadienine, and sphingatrienine. The results of the present study indicate that the diverse activities of Cers reflect structural differences, and this knowledge may pave the way for developing Cer species that are more effective for AD prevention or therapy. Such sphingoid bases could serve as new sphingoid-based chemotherapeutics.

## 4. Materials and Methods

### 4.1. Synthesis of N-((2S,3R,4E,8Z)-1,3-dihydroxyoctadeca-4,8-dien-2-yl-17,17,18,18,18-d_5_)stearamide (Deuterated Plant Cer-d_5_)

(2*S*,3*R*,4*E*,8*Z*)-2-aminooctadeca-4,8-diene-17,17,18,18,18-*d*_5_-1,3-diol (D-*erythro*-(4*E*,8*Z*)-sphingadienine-*d*_5_) was prepared according to our previous report [14]. D-*erythro*-(4*E*,8*Z*)-sphingadienine-*d*_5_ (46.5 mg, 154 μM) and stearic acid (52.5 mg, 185 μM) were dissolved in CH_2_Cl_2_ (5 mL) and cooled to 0 °C. 1-[Bis(dimethylamino)methylene]-1*H*-benzotriazolium 3-oxide hexafluorophosphate (70.1 mg, 185 μM) and *N*,*N*-diisopropylethylamine (80 μL, 450 μM) were added, and the reaction mixture was stirred for 6 h at room temperature. The reaction was quenched with 1 M citric acid, washed with brine, and dried over MgSO_4_. The organic solvent was removed, and the crude product was purified by silica column chromatography (*n*-hexane/EtOAc = 3:1) to obtain deuterated plant Cer-*d*_5_ (47.2 mg, 53%) as white wax. ^1^H NMR (500 MHz, CDCl_3_) δ 6.38 (d, *J* = 7.5 Hz, 1H), 5.81–5.73 (m, 1H), 5.54 (dd, *J* = 15.5, 6.4 Hz, 1H), 5.42–5.28 (m, 2H), 4.30 (brs, 1H), 4.01–3.86 (m, 2H), 3.77–3.66 (m, 1H), 3.38 (brs, 2H), 2.25–2.18 (m, 2H), 2.13–2.09 (m, 2H), 2.09–1.92 (m, 2H), 1.62 (m, 2H), 1.35–1.22 (m, 42H), and 0.87 (t, J = 6.9 Hz, 3H). ^13^C NMR (126 MHz, CDCl_3_) δ 174.20, 133.35, 130.79, 129.26, 128.45, 74.28, 62.33, 54.62, 36.82, 32.37, 31.92, 29.70, 29.67, 29.65, 29.53, 29.38, 29.36, 29.30, 27.32, 26.72, 25.77, 22.68, and 14.10. HRMS (m/z): [M + H]^+^ was calculated for C_36_H_65_D_5_NO_3_, 569.5661; observed, 569.6831.

### 4.2. Preparation of Plant Cers

Plant Cers were prepared according to a published procedure [12]. Briefly, various plant Cers (4*E*,8*Z* d18:2/16 h:0, 4*E*,8*E* d18:2/16 h:0, 4*E*,8*Z* d18:2/18 h:0, 4*E*,8*E* d18:2/18 h:0, 4*E*,8*Z* d18:2/20 h:0, 4*E*,8*E* d18:2/20 h:0, 8*Z* t18:1/22 h:0, 8*Z* t18:1/24 h:0) were prepared from GlcCer (Nagara Science, Gifu, Japan) by glucose hydrolysis with endoglycoceramidase I (EGCase I), isolated from a mutant Rhodococcus erythropolis L-88 strain. Each Cer was extracted by the Bligh–Dyer method and separated by medium-pressure LC on a Hi-Flash S column (Yamazen Corp., Osaka, Japan).

### 4.3. Cell Culture and Treatments

Neuroblastoma SH-SY5Y cells were maintained at 37 °C with 5% CO_2_ in Eagle’s minimum essential medium/Ham’s F-12 medium (ThermoFisher Scientific, Waltham, MA, USA) supplemented with 10% fetal bovine serum (FBS). Primary neuronal cultures were prepared from cerebral cortices of embryonic day 15 mice using a dissociation solution (Sumitomo Bakelite, Tokyo, Japan). The cells were plated on polyethyleneimine-coated dishes and cultured for 7 days in neurobasal medium with 25 mM KCl, 2 mM glutamine, and B27 supplement (ThermoFisher Scientific, Waltham, MA, USA). ReproNeuro, a neuron progenitor derived from human iPSCs, was purchased from ReproCELL (Yokohama, Japan) and maintained in ReproNeuro maturation medium for 14 days according to the manufacturer’s instructions.

The above cells cultured on a 12-well plate were separately treated for 24 h by adding 10 μM plant Cers of different species (4*E*,8*Z* d18:2/16 h:0, 4*E*,8*E* d18:2/16 h:0, 4*E*,8*Z* d18:2/18 h:0, 4*E*,8*E* d18:2/18 h:0, 4*E*,8*Z* d18:2/20 h:0, 4*E*,8*E* d18:2/20 h:0, 8*Z* t18:1/22 h:0, 8*Z* t18:1/24 h:0) or 0.1% dimethylsulfoxide (DMSO; control) to the medium. Cell viability was assessed using a Cell-Counting Kit-8 (Dojindo, Osaka, Japan).

For RNA-mediated interference (RNAi) experiments, Silencer Select siRNAs (ThermoFisher Scientific, Waltham, MA, USA) were used with the following sequences: 5′-CCUACCUGUUUGGUCCUUAtt-3′ (sense) and 5′-UAAGGACCAAACAGGUAGGat-3′ (antisense) for LAPTM4B. Silencer Select Negative Control No. 1 siRNA (ThermoFisher Scientific, Waltham, MA, USA) was used as a control. The siRNAs were delivered with Lipofectamine RNAiMAX reagent (ThermoFisher Scientific, Waltham, MA, USA) according to the manufacturer’s protocol. All experiments were performed 2 days after transfection.

Neuroblastoma SH-SY5Y cells with and without stable transfection of human APP770 were maintained in Eagle’s minimum essential medium/Ham’s F-12 medium (Thermo Scientific, Waltham, MA, USA) supplemented with 1% nonessential amino acids and 10% fetal-bovine serum. The murine microglial BV-2 cell line was purchased from Istituto Nazionale per la Ricerca sul Cancro (Genova, Italy) and cultured in RPMI 1640 (ThermoFisher Scientific, Waltham, MA, USA) supplemented with 10% fetal-bovine serum and L-glutamine. For Transwell cultures, APP770 expressing SH-SY5Y cells (5 × 10^5^/cm^2^) cultured on 24-well plate inserts (0.5 μm pore; Corning, NY, USA) and BV-2 cells (1 × 10^6^) placed below the inserts were treated for 24 h with 10 μM Cers in Eagle’s minimum essential medium/Ham’s F-12 medium. The levels of Aß in the Transwell culture medium were measured with a sandwich ELISA (Wako, Osaka, Japan).

### 4.4. Exosome Collection and Quantification

Exosomes were collected from culture supernatants of SH-SY5Y cells by centrifuging the culture at 2000 g for 10 min and then centrifuging at 10,000 g for 30 min at 4 °C to remove cells and debris. The supernatant was then centrifuged at 100,000 g for 1 h at 4 °C to obtain exosomes as pellets. Exosomes were measured by PS-Capture Exosome ELISA (Fujifilm Wako, Osaka, Japan) according to the manufacturer’s instructions. Antibodies against human CD63 and mouse CD9 (MAB5218, R&D Systems, Minneapolis, MN, USA) were used for SH-SY5Y cell and human iPS neuron-derived EVs and primary mouse neuron-derived EVs, respectively. The size and number of exosomes were measured using a qNano nanoparticle analyzer (Izon Science, Cambridge, MA, USA). CPC100 served as a calibration sample.

### 4.5. Ceramide Overlay Assay

Recombinant human LAPTM4B containing a GST-tag at the N-terminus (Abnova, Taipei, Taiwan) was used in this study. In brief, 10 pmol of ceramides was spotted on nitrocellulose membranes. After the blocking with Blocking One (Nacalai tesque, Kyoto, Japan), the membranes were incubated with 1 nM LAPTM4B overnight at 4 °C. LAPTM4B bound to the ceramides on the membrane was detected using an HRP-conjugated anti-GST antibody. The dot densities were quantified using ImageJ software 2.0.0 (Bethesda, MD, USA).

### 4.6. Deuterated Plant Cer-d5 Extraction and Quantification by LC-MS/MS

Each deuterated plant Cer-*d*_5_ (4*E*,8*Z* d18:2-*d*_5_/18:0) treated SH-SY5Y cell and exosome sample was gently sonicated in ice-cold PBS (0.5–1 mL). Total lipids in cells and exosomes were extracted with chloroform/methanol (2:1 *v*/*v*; 3 mL) at 48 °C for 2 h with 0.1 nmol of internal standard sphingolipid (d18:1/17:0). Glycerolipids were saponified by 10 M NaOH in water (0.1 mL) and incubated at 37 °C for 2 h. Chloroform (1.5 mL) and water (0.5 mL) were added and centrifuged to generate two phases. The hydrophobic phase was collected and analyzed by LC-MS/MS using a TripleTOF 5600 System (AB SCIEX, Foster City, CA, USA) equipped with an electrospray ionization (ESI) probe and interfaced with a Prominence UFLC system (Shimadzu, Kyoto, Japan) according to our previous report [14,22]. Extracted lipids (5 μL) were injected onto an InertSustain NH2 column (particle size 5 μm, diameter 2.1 mm, length 100 mm; GL Science, Tokyo, Japan). Mobile phases A and B consisted of acetonitrile/methanol/formic acid = 95:5:0.2 (*v*/*v*) containing 5 mM ammonium formate and methanol/formic acid = 100:0.2 (*v*/*v*) containing 5 mM ammonium formate, respectively. Lipids were eluted at 0.13 mL/min by applying 45 min solvent gradient consisting of 0–5 min 0% B, 5–10 min 0% to 20% B, 10–12 min 20% B, 12–15 min 20% to 50% B, 15–22 min 50% B, 22–27 min 50% to 80% B, 27–30 min 80% B, and 30–45 min 80% to 0% B. LC-MS/MS analysis was conducted in positive ion mode with internal standards d18:1/17:0 ([M + H]^+^ *m*/*z* = 552.5) and 4*E*,8*Z* d18:2-*d*_5_/18:0 ([M + H]^+^ *m*/*z* = 569.5) as precursor ions. Analytes d18:1-(sphingosine) and 4*E*,8*Z* d18:2-*d*_5_-bound Cer were identified from their retention times and characteristic product ions (*m*/*z* = 264.2691 for d18:1-bound Cer, 267.2849 for 4*E*,8*Z* d18:2-*d*_5_-bound Cer). Proteins were quantified using 50 μL of cell lysates and a BCA assay kit (Nacalai Tesque, Kyoto, Japan) according to the manufacturer’s protocol, and lipid amounts were normalized by the protein content of the cells.

## Figures and Tables

**Figure 1 ijms-23-10751-f001:**
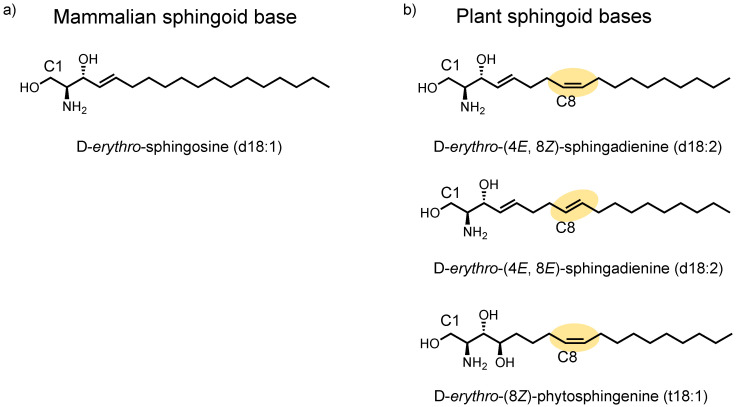
Structures of the mammalian sphingoid base sphingosine (**a**) and plant sphingoid bases sphingadienine and phytosphingenine (**b**).

**Figure 2 ijms-23-10751-f002:**
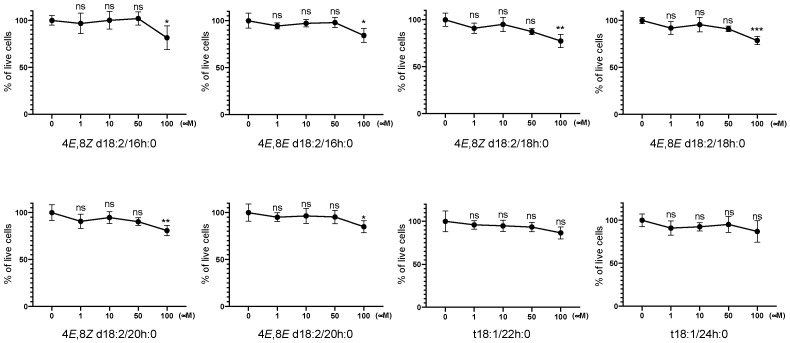
Cell Counting Kit-8 (CCK-8) cell viability assays of SH-SY5Y cells exposed to typical plant Cers. Results were normalized against controls and were represented as the mean ± standard deviation (SD; *n* = 4; * *p* < 0.05, ** *p* < 0.01, *** *p* < 0.001 vs. controls by *t*-test).

**Figure 3 ijms-23-10751-f003:**
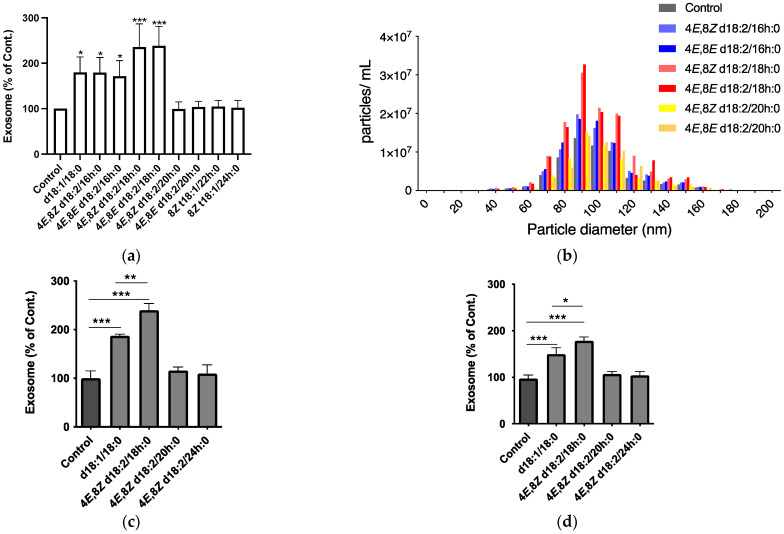
(**a**) Quantification of exosomes released from SH-SY5Y cells treated without (Control) or with typical plant Cers for 24 h. Exosomes were quantified using a PS-Capture Exosome ELISA system. Results are normalized against controls and represented as the mean ± SD (*n* = 3; * *p* < 0.05, *** *p* < 0.0001 vs. Control). (**b**) Particle size distribution of exosomes derived from control and Cer-treated cells. (**c**,**d**) Quantification of exosomes released from primary mouse neurons (**c**) and human iPS neurons treated without (Control) or with the indicated Cers for 24 h (**d**). Exosomes were measured using a PS-Capture Exosome ELISA system. Results were normalized against controls and represented as the mean ± SD (*n* = 3; * *p* < 0.05, ** *p* < 0.01, *** *p* < 0.001 by one-way ANOVA).

**Figure 4 ijms-23-10751-f004:**
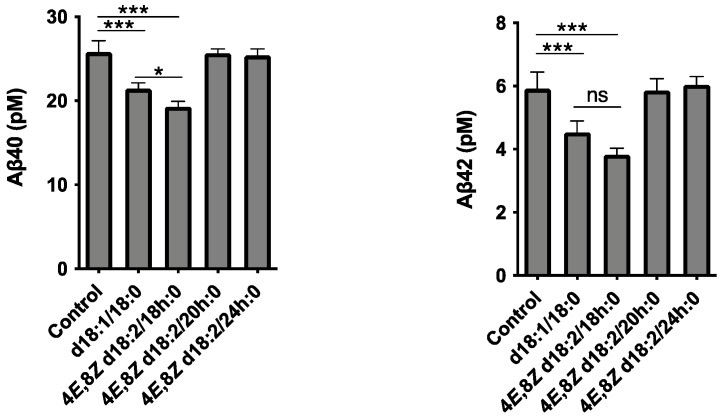
The levels of Aβ40 and Aß42 in medium from Transwell cultured APP-expressing SH-SY5Y and BV-2 cells. After 24 h of the indicated Cers treatment, Aβ levels in culture media were determined. Values are the means ± SD. (*n* = 3, * *p* < 0.05; *** *p* < 0.001 by one-way ANOVA).

**Figure 5 ijms-23-10751-f005:**
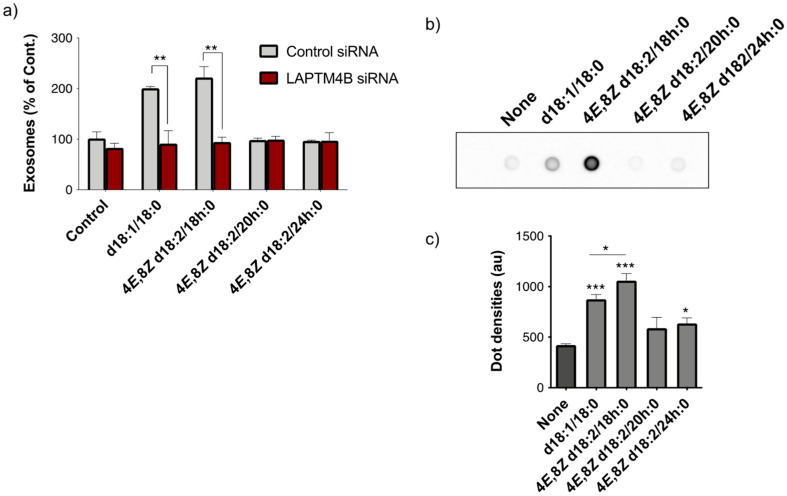
(**a**) The numbers of exosomes in the conditioned medium of SH-SY5Y cells treated with or without plant Cers following siRNA-induced knockdown of LAPTM4B. Data were normalized to the control and are represented as the mean ± S.D. (*n* = 4). ** *p* < 0.01 (versus the Control siRNA). (**b**) A representative image of protein-ceramide overlay assay using GST-fused LAPTM4B and the indicated Cers. (**c**) The signal intensity of each dot was analyzed, and the data were represented as the mean ± S.D. (*n* = 3; * *p* < 0.05, ** *p* < 0.01, *** *p* < 0.001, versus Control). au; arbitrary unit.

**Figure 6 ijms-23-10751-f006:**
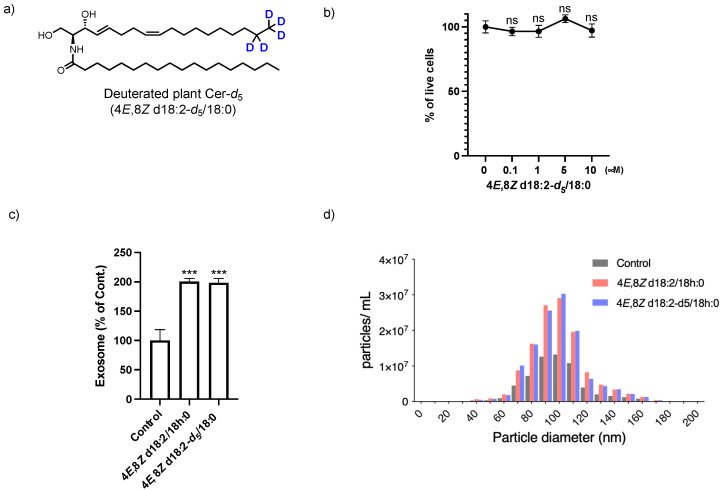
(**a**) Structure of 4*E*,8*Z* d18:2-*d*_5_/18:0. (**b**) CCK-8 viability assay of SH-SY5Y cells exposed to 4*E*,8*Z* d18:2-*d*_5_/18:0. Results were normalized against controls and represented as the mean ± SD (*n* = 3). (**c**) Quantification of exosomes released from SH-SY5Y cells treated without (Control) or with 4*E*,8*Z* d18:2-*d*_5_/18:0 for 24 h. Exosomes were quantified using a PS-Capture Exosome ELISA system. Results were normalized against controls and represented as the mean ± SD (*n* = 3; *** *p* < 0.0001 vs. Control). (**d**) Particle size distributions of exosomes derived from control, and deuterium-labeled or unlabeled Cer-treated cells.

**Figure 7 ijms-23-10751-f007:**
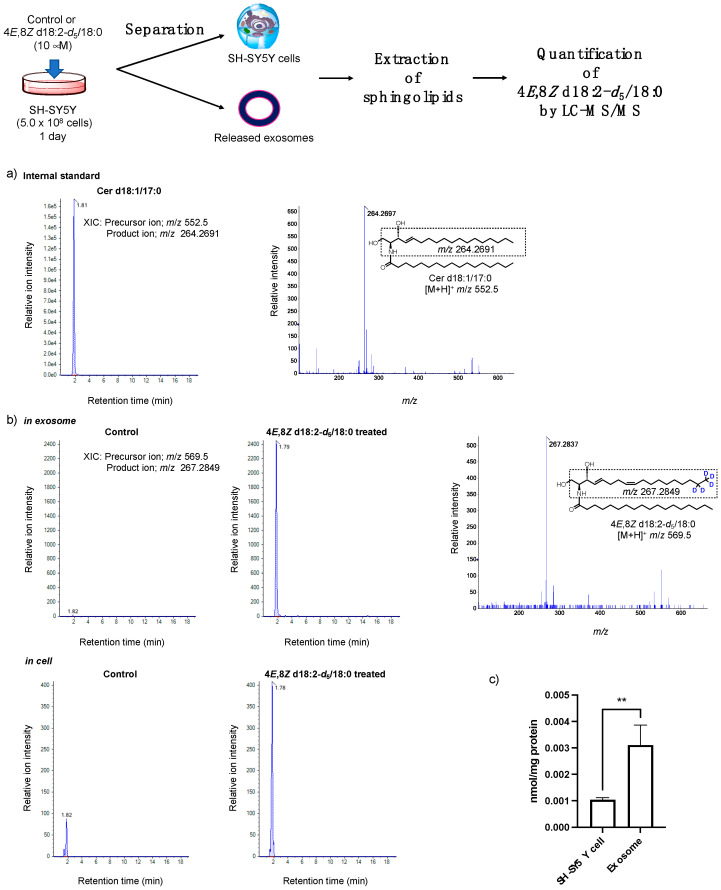
(**a**) Extracted ion chromatogram (XIC) and product ion spectrum for the product ion (*m*/*z* = 264.2691) of Cer d18:1/17:0 (*m*/*z* = 552.5). (**b**) XIC and product ion spectrum of Cer 4*E*,8*Z* d18:2-*d*_5_/18:0 (*m*/*z* = 569.5) clearly showing the product ion *m*/*z* = 267.2849 (4*E*,8*Z* d18:2-*d*_5_) in exosome. (**c**) Quantities of 4*E*,8*Z* d18:2-*d*_5_/18:0 in SH-SY5Y cells and exosomes treated for 24 h. Results were represented as the mean ± SD (*n* = 3; ** *p* < 0.01).

## Data Availability

Not applicable.

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
