# Peer review of "Evaluation of Plant Ceramide Species-Induced Exosome Release from Neuronal Cells and Exosome Loading Using Deuterium Chemistry"

_ijms, 2022, doi:10.3390/ijms231810751_

Round 1
Reviewer 1 Report (Previous Reviewer 1)
Thank you for responding to my comment and correcting the paper.
Reviewer 2 Report (Previous Reviewer 2)
The authors have addressed my concerns in the revised manuscript and I recommend the paper for publication.
This manuscript is a resubmission of an earlier submission. The following is a list of the peer review reports and author responses from that submission.
Round 1
Reviewer 1 Report
This manuscript describes the research on the exosome release by plant ceramide species, which is very interesting, but I got the impression that the discussion was insufficient. For example, I think it is better to consider the description "Cer species with long-chain fatty acids (C16 and C18) strongly promoted exosome release, these with very long-chain fatty acids (C20, C22, and C24) did not exert an effect." including a little more mechanism. I think it is desirable to clearly describe the consideration of the results of this experiment, including other parts.
Reviewer 2 Report
Alzheimer’s disease (AD) is an age-associated neurological disorder that affects memory, thinking and behavior. Extracellular accumulation of aggregated amyloid-b in the brain is a hallmark of the AD pathology. In the manuscript titled “evaluation of plant ceramide species-induced exosome release from neuronal cells and exosome loading using deuterium chemistry”, the authors investigated plant ceramide-induced exosome release and demonstrated that plant ceramides with long chain fatty acids can effectively induce exosome release in SH-SY5Y cells. However, the link between exosome release and Ab clearance/AD pathology is unclear. I’ve detailed my comments below.
1. Although the authors demonstrated ceramide-dependent upregulation of exosome release, it is unclear whether increased exosome release promote Ab clearance. In addition, the authors examined this phenotype in the neuroblastoma SH-SY5Y cell line, it is unclear whether ceramides could upregulate exosome release in neurons. The authors should repeat a few key experiments using primary human/mouse neuron cultures.
2. Can the authors use a mouse model or a cellular model of AD to show that plant ceramides with long chain fatty acids indeed improve Ab clearance?
3. In the cellular experiments examining exosome release in SH-SY5Y cells, the authors carried out experiments only after acute treatment (1 day) with plant ceramides. Can plant ceramides induce exosome release in long term treatment conditions (for example, one week)?
Reviewer 3 Report
The authors previously have reported that plant derived-ceramides can influence the release of exosomes in neuronal cells and can promote the clearance of the AB. In the present study, authors have compared the efficacy of plant and animal derived ceramides on exosome release. However, the authors need to address the following concerns to improve the quality of the manuscript.
1. The introduction is almost repetition of the previously published paper (Sci Rep. 2019; 9: 16827) and the discussion is barely supported by the current findings. The importance of current finding is missing and questionable.
2. The current results only add a little knowledge to the previous findings (Sci Rep. 2019; 9: 16827) and therefore, the novelty of the manuscript is a major concern.
3. Authors have performed no in-vivo experiments to support the conclusion that plant derived ceramides are more effective in prevention or treatment of AD compared to mammal-derived ceramides.
4. Authors should investigate the mechanism of enhanced exosome release by plant ceramides compared to animal derived ceramides to support their claim and improve the quality of the manuscript.